# Speak, Memory: An Archaeology of Books Known to ChatGPT/GPT-4

**Kent K. Chang**
University of California, Berkeley
kentkchang@berkeley.edu

**Mackenzie Cramer**
University of California, Berkeley
mackenzie.hanh@berkeley.edu

**Sandeep Soni**
Emory University
sandeep.soni@emory.edu

**David Bamman**[*]
University of California, Berkeley
dbamman@berkeley.edu

## Abstract

In this work, we carry out a data archaeology to infer books that are known to ChatGPT and GPT-4 using a *name cloze* membership inference query. We find that OpenAI models have memorized a wide collection of copyrighted materials, and that the degree of memorization is tied to the frequency with which passages of those books appear on the web. The ability of these models to memorize an unknown set of books complicates assessments of measurement validity for cultural analytics by contaminating test data; we show that models perform much better on memorized books than on non-memorized books for downstream tasks. We argue that this supports a case for open models whose training data is known.

## 1 Introduction

Research in cultural analytics at the intersection of NLP and narrative is often focused on developing algorithmic devices to measure some phenomenon of interest in literary texts (Piper et al., 2021; Yoder et al., 2021; Coll Ardanuy et al., 2020; Evans and Wilkens, 2018). The rise of large-pretrained language models such as ChatGPT and GPT-4 has the potential to radically transform this space by both reducing the need for large-scale training data for new tasks and lowering the technical barrier to entry (Underwood, 2023).

At the same time, however, these models also present a challenge for establishing the validity of results, since few details are known about the data used to train them. As others have shown, the accuracy of such models is strongly dependent on the frequency with which a model has seen information in the training data, calling into question their ability to generalize (Razeghi et al., 2022; Kandpal et al., 2022a; Elazar et al., 2022); in addition, this phenomenon is exacerbated for larger models (Carlini et al., 2022; Biderman et al., 2023). Knowing

---

[*]Details of author contributions listed in the appendix.

> Wow. I sit down, fish the questions from my backpack, and go through them, inwardly cursing [MASK] for not providing me with a brief biography. I know nothing about this man I'm about to interview. He could be ninety or he could be thirty. → **Kate** (James, *Fifty Shades of Grey*).
>
> Some days later, when the land had been moistened by two or three heavy rains, [MASK] and his family went to the farm with baskets of seed-yams, their hoes and machetes, and the planting began. → **Okonkwo** (Achebe, *Things Fall Apart*).

Figure 1: Name cloze examples. GPT-4 answers both of these correctly.

what books a model has been trained on is critical to assess such sources of bias (Gebru et al., 2021), which can impact the validity of results in cultural analytics: if evaluation datasets contain memorized books, they provide a false measure of future performance on non-memorized books; without knowing what books a model has been trained on, we are unable to construct evaluation benchmarks that can be sure to exclude them.

In this work, we carry out a *data archaeology* to infer books that are known to several of these large language models. This archaeology is a membership inference query (Shokri et al., 2017) in which we probe the degree of exact memorization (Tirumala et al., 2022) for a sample of passages from 571 works of fiction published between 1749–2020. This difficult name cloze task, illustrated in figure 1, has 0% human baseline performance.

This archaeology allows us to uncover a number of findings about the books known to OpenAI models which can impact downstream work in cultural analytics:

1. OpenAI models, and GPT-4 in particular, have

memorized a wide collection of in-copyright books.

2. While others have shown that LLMs are able to reproduce some popular works (Henderson et al., 2023), we measure the systematic biases in what books OpenAI models have seen and memorized, with the most strongly memorized books including science fiction/fantasy novels, popular works in the public domain, and bestsellers.

3. This bias aligns with that present in the general web, as reflected in search results from Google, Bing and C4. This confirms prior findings that duplication encourages memorization (Carlini et al., 2023), and provides a rough diagnostic for assessing knowledge about a book.

4. Disparity in memorization leads to disparity in downstream tasks. GPT models perform better on memorized books than non-memorized books at predicting the year of first publication for a work and the duration of narrative time for a passage, and are more likely to generate character names from books it has seen.

While our work is focused on ChatGPT and GPT-4, we also uncover surprising findings about BERT as well: BookCorpus (Zhu et al., 2015), one of BERT's training sources, contains in-copyright materials by published authors, including E.L. James' *Fifty Shades of Grey*, Diana Gabaldon's *Outlander* and Dan Brown's *The Lost Symbol*, and BERT has memorized this material as well.

As researchers in cultural analytics are poised to use ChatGPT and GPT-4 for the empirical analysis of literature, our work both sheds light on the underlying knowledge in these models while also illustrating the threats to validity in using them.

## 2   Related Work

**Knowledge production and critical digital humanities.**   The archaeology of data that this work presents can be situated in the tradition of tool critiques in critical digital humanities (Berry, 2011; Fitzpatrick, 2012; Hayles, 2012; Ramsay and Rockwell, 2012; Ruth et al., 2022), where we critically approach the closedness and opacity of LLMs, which can pose significant issues if they are used to reason about literature (Elkins and Chun, 2020; Goodlad and Dimock, 2021; Henrickson and Meroño-Peñuela, 2022; Schmidgen et al., 2023;

Elam, 2023). In this light, our archaeology of books shares the Foucauldian impulse to "find a common structure that underlies both the scientific knowledge and the institutional practices of an age" (Gutting, 1989, p. 79; Schmidgen et al., 2023). to the inference tasks that involve them. Investigating data membership helps us reflect on how to best use LLMs for large-scale cultural analysis and evaluate the validity of its findings.

**LLMs for cultural analysis.**   While they are more often the object of critique, LLMs are gaining prominence in large-scale cultural analysis as part of the methodology. For example, some use GPT models to tackle problems related to interpretation, that of real events (Hamilton and Piper, 2022), or of character roles (hero, villain, and victim, Stammbach et al., 2022). Others leverage LLMs on classic NLP tasks that can be used to shed light on literary and cultural phenomena, or otherwise maintain an interest in those humanistic domains (Ziems et al., 2023). This work shows what tasks and what types of questions LLMs are more suitable to answer than others through our archaeology of data.

**Documenting training data.**   Training on large text corpora such as BookCorpus (Zhu et al., 2015), C4 (Raffel et al., 2020) and the Pile (Gao et al., 2020) has been instrumental in extending the capability of large language models. Yet, in contrast to smaller datasets, these large corpora are less carefully curated. Besides a few attempts at documenting large datasets (e.g., C4; Dodge et al., 2021) or laying out diagnostic techniques for their quality (Swayamdipta et al., 2020), these corpora and their use in large models is less understood. Our focus on books in this work is an attempt to empirically map the information about books that is present in these models.

**Memorization.**   Large language models have shown impressive zero-shot or few-shot ability but they also suffer from memorization (Elangovan et al., 2021; Lewis et al., 2021). While memorization is shown in some cases to improve generalization (Khandelwal et al., 2020), it has generally been shown to have negative consequences, including security and privacy risks (Carlini et al., 2021, 2023; Huang et al., 2022). Studies have quantified the level of memorization in large language models (e.g., Carlini et al., 2022; Mireshghallah et al., 2022) and have highlighted the role of both *verbatim* (e.g., Lee et al., 2022; Ippolito et al., 2022)

> You have seen the following passage in your training data. What is the proper name that fills in the [MASK] token in it? This name is exactly one word long, and is a proper name (not a pronoun or any other word). You must make a guess, even if you are uncertain.
>
> Example:
>
> Input: Stay gold, [MASK], stay gold.
> Output: <name>Ponyboy</name>
>
> Input: The door opened, and [MASK], dressed and hatted, entered with a cup of tea.
> Output: <name>Gerty</name>
>
> Input: My back's to the window. I expect a stranger, but it's [MASK] who pushes open the door, flicks on the light. I can't place that, unless he's one of them. There was always that possibility.
> Output:

Figure 2: Sample name cloze prompt.

and subtler forms of memorization (Zhang et al., 2021). Henderson et al. (2023) in particular examines the legal issues surrounding copyright in foundation models, focusing in particular on the length of content extraction of copyrighted materials (including books as a case study) that poses the most risk to fair use determinations. Our analysis and experimental findings on the disparate sources of memorization and impact on research in cultural analytics add to this scholarship.

**Data contamination.** A related issue noted upon critical scrutiny of uncurated large corpora is data contamination (e.g., Magar and Schwartz, 2022) raising questions about the zero-shot capability of these models (e.g., Blevins and Zettlemoyer, 2022) and worries about security and privacy (Carlini et al., 2021, 2023). For example, Dodge et al. (2021) find that text from NLP evaluation datasets is present in C4, attributing performance gain partly to the train-test leakage. Lee et al. (2022) show that C4 contains repeated long running sentences and near duplicates whose removal mitigates data contamination; similarly, deduplication has also been shown to alleviate the privacy risks in models trained on large corpora (Kandpal et al., 2022b).

## 3 Task

### 3.1 Name cloze

We formulate our task as a cloze: given some context, predict a single token that fills in a mask. To account for different texts being more predictable than others, we focus on a hard setting of predicting the identity of a single *name* in a passage of 40–60 tokens that contains no other named entities. Figure 1 illustrates two such examples.

In the following, we refer to this task as a *name*

*cloze*, a version of *exact memorization* (Tirumala et al., 2022). Unlike other cloze tasks that focus on entity prediction for question answering/reading comprehension (Hill et al., 2015; Onishi et al., 2016), no names at all appear in the context to inform the cloze fill. In the absence of information about each particular book, this name should be nearly impossible to predict from the context alone; it requires knowledge not of English, but rather about the work in question.

The name cloze setup is also particularly suitable for our data archaeology for ChatGPT/GPT-4, as opposed to perplexity-based measures (Gonen et al., 2022), since OpenAI does not at the time of writing share word probabilities through their API.

### 3.2 Evaluation

We construct this evaluation set by running BookNLP[1] over the dataset described below in §4, extracting all passages between 40 and 60 tokens with a single proper person entity and no other named entities. Each passage contains complete sentences, and does not cross sentence boundaries. We randomly sample 100 such passages per book, and exclude any books from our analyses with fewer than 100 such passages.

We pass each passage through the prompt listed in figure 2, which is designed to elicit a single word, proper name response wrapped in XML tags; two short input/output examples are provided to illustrate the expected structure of the response.

In establishing baselines using the same evaluation set, predicting the most frequent name in the dataset ("Mary") yields an accuracy of 0.6%. To assess human performance on this task, one of the authors of this paper followed the same instruc-

---

[1] https://github.com/booknlp/booknlp

tions given ChatGPT/GPT-4, only using information present in the passage to guess the masked name (i.e., without using any external sources of information such as Google); the resulting accuracy is 0%, which suggests there is little information within the context that would provide a signal for what the true name should be.

## 4 Data

We evaluate 5 sources of English-language fiction:

- 91 novels from LitBank, published before 1923.

- 90 Pulitzer prize nominees from 1924–2020.

- 95 Bestsellers from the *NY Times* and *Publishers Weekly* from 1924–2020.

- 101 novels written by Black authors, either from the Black Book Interactive Project[2] or Black Caucus American Library Association award winners from 1928–2018.

- 95 works of Global Anglophone fiction (outside the U.S. and U.K.) from 1935–2020.

- 99 works of genre fiction, containing science fiction/fantasy, horror, mystery/crime, romance and action/spy novels from 1928–2017.

Pre-1923 LitBank texts are born digital on Project Gutenberg and are in the public domain in the United States; all other sources were created by purchasing physical books, scanning them and OCR'ing them with Abbyy FineReader. As of the time of writing, books published after 1928 are generally in copyright in the U.S.

## 5 Results

### 5.1 ChatGPT/GPT-4

We pass all passages with the same prompt through both ChatGPT (`gpt-3.5-turbo`) and GPT-4 (`gpt-4`), using the OpenAI API. The total cost of this experiment with current OpenAI pricing ($0.002/thousand tokens for ChatGPT; $0.03/thousand tokens for GPT-4) is approximately $400. We measure the name cloze accuracy for a book as the fraction of 100 samples from it where the model being tested predicts the masked name correctly.

Table 1 presents the top 20 books with the highest GPT-4 name cloze accuracy. While works in the public domain dominate this list, table 6 in the

Appendix presents the same for books published after 1928.[3] Of particular interest in this list is the dominance of science fiction and fantasy works, including *Harry Potter*, *1984*, *Lord of the Rings*, *Hunger Games*, *Hitchhiker's Guide to the Galaxy*, *Fahrenheit 451*, *A Game of Thrones*, and *Dune*— 12 of the top 20 most memorized books in copyright fall in this category. Table 2 explores this in more detail by aggregating the performance by the top-level categories described above, including the specific genre for our subset of genre fiction.

GPT-4 and ChatGPT are widely knowledgeable about texts in the public domain (included in pre-1923 LitBank); it knows little about works of Global Anglophone texts, works in the Black Book Interactive Project and Black Caucus American Library Association award winners.

| Source | GPT-4 | ChatGPT |
|---|---|---|
| pre-1923 LitBank | 0.244 | 0.072 |
| Genre: SF/Fantasy | 0.235 | 0.108 |
| Genre: Horror | 0.054 | 0.028 |
| Bestsellers | 0.033 | 0.016 |
| Genre: Action/Spy | 0.032 | 0.007 |
| Genre: Mystery/Crime | 0.029 | 0.014 |
| Genre: Romance | 0.029 | 0.011 |
| Pulitzer | 0.026 | 0.011 |
| Global | 0.020 | 0.009 |
| BBIP/BCALA | 0.017 | 0.011 |

Table 2: Name cloze performance by book category.

### 5.2 BERT

For comparison, we also generate predictions for the masked token using BERT (only passing the passage through the model and not the prefaced instructions) to provide a baseline for how often a model would guess the correct name when simply functioning as a language model (unconstrained to generate proper names). As table 1 illustrates, BERT's performance is near 0 for all books—except for *Fifty Shades of Grey*, for which it guesses the correct name 13% of the time, suggesting that this book was known to BERT during training. Devlin et al. (2019) notes that BERT was trained on Wikipedia and the BookCorpus, which Zhu et al. (2015) describe as "free books written by yet unpublished authors."[4] Manual inspection of the BookCorpus hosted by huggingface[5] confirms

---

[2]http://bbip.ku.edu/novel-collections

[3]For complete results on all books, see https://github.com/bamman-group/gpt4-books.

[4]*Fifty Shades of Grey* was originally self-published on fanfiction.net ca. 2009 before being published by Vintage Books in 2012.

[5]https://huggingface.co/datasets/bookcorpus

| GPT-4 | ChatGPT | BERT | Date | Author | Title |
|-------|---------|------|------|--------|-------|
| 0.98 | 0.82 | 0.00 | 1865 | Lewis Carroll | *Alice's Adventures in Wonderland* |
| 0.76 | 0.43 | 0.00 | 1997 | J.K. Rowling | *Harry Potter and the Sorcerer's Stone* |
| 0.74 | 0.29 | 0.00 | 1850 | Nathaniel Hawthorne | *The Scarlet Letter* |
| 0.72 | 0.11 | 0.00 | 1892 | Arthur Conan Doyle | *The Adventures of Sherlock Holmes* |
| 0.70 | 0.10 | 0.00 | 1815 | Jane Austen | *Emma* |
| 0.65 | 0.19 | 0.00 | 1823 | Mary W. Shelley | *Frankenstein* |
| 0.62 | 0.13 | 0.00 | 1813 | Jane Austen | *Pride and Prejudice* |
| 0.61 | 0.35 | 0.00 | 1884 | Mark Twain | *Adventures of Huckleberry Finn* |
| 0.61 | 0.30 | 0.00 | 1853 | Herman Melville | *Bartleby, the Scrivener* |
| 0.61 | 0.08 | 0.00 | 1897 | Bram Stoker | *Dracula* |
| 0.61 | 0.18 | 0.00 | 1838 | Charles Dickens | *Oliver Twist* |
| 0.59 | 0.13 | 0.00 | 1902 | Arthur Conan Doyle | *The Hound of the Baskervilles* |
| 0.59 | 0.22 | 0.00 | 1851 | Herman Melville | *Moby Dick; Or, The Whale* |
| 0.58 | 0.35 | 0.00 | 1876 | Mark Twain | *The Adventures of Tom Sawyer* |
| 0.57 | 0.30 | 0.00 | 1949 | George Orwell | *1984* |
| 0.54 | 0.10 | 0.00 | 1908 | L. M. Montgomery | *Anne of Green Gables* |
| 0.51 | 0.20 | 0.01 | 1954 | J.R.R. Tolkien | *The Fellowship of the Ring* |
| 0.49 | 0.16 | 0.13 | 2012 | E.L. James | *Fifty Shades of Grey* |
| 0.49 | 0.24 | 0.01 | 1911 | Frances H. Burnett | *The Secret Garden* |
| 0.49 | 0.12 | 0.00 | 1883 | Robert L. Stevenson | *Treasure Island* |
| 0.49 | 0.16 | 0.00 | 1847 | Charlotte Brontë | *Jane Eyre: An Autobiography* |
| 0.49 | 0.22 | 0.00 | 1903 | Jack London | *The Call of the Wild* |

Table 1: Top 20 books by GPT-4 name cloze accuracy.

that *Fifty Shades of Grey* is present within it, along with several other published works, including Diana Gabaldon's *Outlander* and Dan Brown's *The Lost Symbol*.

## 6  Analysis

### 6.1  Error analysis

We analyze examples on which ChatGPT and GPT-4 make errors to assess the impact of memorization. Specifically, we test the following question: when a model makes a name cloze error, is it more likely to offer a name from a memorized book than a non-memorized one?

To test this, we construct sets of seen ($S$) and unseen ($U$) character names by the models. To do this, we divide all books into three categories: $M$ as books the model has memorized (top decile by GPT-4 name cloze accuracy), $\neg M$ as books the model has not memorized (bottom decile), and $H$ as books held out to test the hypothesis. We identify the true masked names that are most associated with the books in $M$ — by calculating the positive pointwise mutual information between a name and book pair — to obtain set $S$, and the masked names most associated with books in $\neg M$ to obtain set $U$. We also ensure that $S$ and $U$ are of the same size and have no overlap. Next, we calculate the observed statistic as the log-odds ratio on examples

from $H$:

$$o = \log \left( \frac{Pr\left(\hat{c} \in S | error\right)}{Pr\left(\hat{c} \in U | error\right)} \right),$$

where $\hat{c}$ is the predicted character. To test for statistical significance, we perform a randomization test (Dror et al., 2018), where the observed statistic $o$ is compared to a distribution of the same statistic calculated by randomly shuffling the names between $S$ and $U$.

We find that for both ChatGPT ($o = 1.34, p < 0.0001$) and GPT-4 ($o = 1.37, p < 0.0001$), the null hypothesis can be rejected, indicating that both models are more likely to predict a character name from a book they had memorized than a character from a book they have not. This has important consequences: these models do not simply perform better on a set of memorized books, but the information from those books bleeds out into other narrative contexts.

### 6.2  Extrinsic analysis

Why do the GPT models know about some books more than others? As others have shown, duplicated content in training is strongly correlated with memorization (Carlini et al., 2023) and foundation models are able to reproduce popular texts more so than random ones (Henderson et al., 2023). While the training data for ChatGPT and GPT-4 is unknown, it likely involves data scraped from the web, as with prior models' use of WebText and C4.

To what degree is a model's performance for a book in our name cloze task correlated with the number of copies of that book on the open web? We assess this using four sources: Google and Bing search engine results, C4, and the Pile.

For each book in our dataset, we sample 10 passages at random from our evaluation set and select a 10-gram from it; we then query each platform to find the number of search results that match that exact string. We use the custom search API for Google,[6] the Bing Web Search API[7] and indexes to C4 and the Pile by AI2 (Dodge et al., 2021).[8]

Table 3 lists the results of this analysis, displaying the correlation (Spearman $\rho$) between GPT-4 name cloze accuracy for a book and the average number of search results for all query passages from it.

| Date | Google | Bing | C4 | Pile |
|------|--------|------|------|------|
| pre-1928 | 0.74 | 0.70 | 0.71 | 0.84 |
| post-1928 | 0.37 | 0.41 | 0.36 | 0.21 |

Table 3: Correlation (Spearman $\rho$) between GPT-4 name cloze accuracy and number of search results in Google, Bing, C4 and the Pile.

For works in the public domain (published before 1928), we see a strong and significant ($p < 0.001$) correlation between GPT-4 name cloze accuracy and the number of search results across all sources. Google, for instance, contains an average of 2,590 results for 10-grams from *Alice in Wonderland*, 1,100 results from *Huckleberry Finn* and 279 results from *Anne of Green Gables*. Works in copyright (published after 1928) show up frequently on the web as well. While the correlation is not as strong as public domain texts (in part reflecting the smaller number of copies), it is strongly significant as well ($p < 0.001$). Google again has an average of 3,074 search results across the ten 10-ngrams we query for *Harry Potter and the Sorcerer's Stone*,[9] 92 results for *The Hunger Games* and 41 results for *A Game of Thrones*.

While this does not indicate direct causation between web prevalence and memorization, we speculate that there is a hidden confounder that explains

them both: popularity. *Moby Dick*, for example, is widely read: many copies of it appear in university libraries, multiple copies of it may appear in large-scale training datasets (such as Books3, part of Pile), quotations of it may appear in unindexed academic journals, and as we see here, it appears in multiple places on the public web (both in snippets and in the form of full text). All four of these factors are likely correlated with each other and commonly explained by the overall popularity of the text itself (concretely, by an unobservable quantity such as the number of times it has been read worldwide). The correlation we see between web prevalence and memorization may be an indirect reflection of this latent causal graph.

| Domain | Hits |
|--------|------|
| archive.org | 337 |
| academia.edu | 257 |
| goodreads.com | 234 |
| coursehero.com | 197 |
| quizlet.com | 181 |
| litcharts.com | 148 |
| fliphtml5.com | 124 |
| genius.com | 118 |
| amazon.com | 109 |
| issuu.com | 98 |

Table 4: Sources for copyrighted material.

Table 4 lists the most popular sources for copyrighted material. Notably, this list does not only include sources where the full text is freely available for download as a single document, but also sources where smaller snippets of the texts appear in reviews (on Goodreads and Amazon) and in study resources (such as notes and quizzes on CourseHero, Quizlet and LitCharts). Since our querying process samples ten random passages from a book (not filtered according to notability or popularity in any way), this suggests that there exists a significant portion of copyrighted literary works in the form of short quotes or excerpts on the open web. On Goodreads, for example, a book can have a dedicated page of quotes from it added by users of the website (in addition to quotes highlighted in their reviews; cf. Walsh and Antoniak, 2021); on LitCharts, the study guide of a book often includes a detailed chapter-by-chapter summary that includes quotations.

In addition, this also suggests that books that are reviewed and maintain an online presence or assigned as course readings are more likely to appear on the open web, despite copyright restrictions. In

---

[6]https://developers.google.com/custom-search/v1/overview
[7]https://www.microsoft.com/en-us/bing/apis/bing-web-search-api
[8]https://c4-search.apps.allenai.org
[9]cf. "seemed to know where he was going, he was obviously"; "He bent and pulled the ring of the trapdoor, which"; "I got a few extra books for background reading, and"

this light, observations from literary scholars and sociologies remain pertinent: Bourdieu's (1987) concerns, that literary taste reflects and reinforces social inequalities, could also be relevant in the context of LLMs. Guillory's (1993)'s critique on canon formation, which highlights the role of educational institutions, can still serve as a potent reminder of the impact of literary syllabi: since they have influence over what books should have flashcards and study guides created for them, they also indirectly sway which books might become more prominent on the open web, and likely in the training data for LLMs.

## 7 Effect on downstream tasks

The varying degrees to which ChatGPT and GPT-4 have seen books in their training data has the potential to bias the accuracy of downstream analyses. To test the degree to which this is true, we carry out two predictive experiments using GPT-4 in a few-shot scenario: predicting the year of first publication of a work from a 250-word passage, and predicting the amount of time that has passed within it. Note that here we are interested in the effects of disparity in memorization on downstream tasks, not any individual model's performance on them, and for this reason, we focus on a few-shot setup without any fine-tuning.

### 7.1 Predicting year of first publication

Our first experiment predicts the year of first publication of a book from a random 250-word passage of text within it, a task widely used in the digital humanities when the date of *composition* for a work is unknown or contested (Jong et al., 2005; Kumar et al., 2011; Bamman et al., 2017; Karsdorp and Fonteyn, 2019).

For each of the books in our dataset, we sample a random 250-word passage (with complete sentences) and pass it through the prompt shown in figure 3 in the Appendix to solicit a four-digit date as a prediction for the year of first publication. We measure two quantities: whether a model is able make a valid year prediction at all (or refrains, noting the year is "uncertain"); if it make a valid prediction, we calculate the prediction error for each book as the absolute difference between the predicted year and the true year ($|\hat{y} - y|$). In order to assess any disparities in performance as a function of a model's knowledge about a book, we focus on the differential between the books in the

top 10% and bottom 10% by GPT-4 name cloze accuracy.

As table 5 shows, we see strong disparities in performance as result of a model's familiarity with a book. ChatGPT refrains from making a prediction in only 4% of books it knows well, but 9% in books it does not (GPT-4 makes a valid prediction in all cases). When ChatGPT makes a valid prediction, the mean absolute error is much greater for the books it has not memorized (29.8 years) than for books it has (3.3 years); and the same holds for GPT-4 as well (14.5 years for non-memorized books; 0.6 years for memorized ones). Over all books with predictions, increasing GPT name cloze accuracy is linked to decreasing error rates at year prediction (Spearman $\rho = -0.39$ for ChatGPT, $-0.37$ for GPT-4, $p < 0.001$).

Predicting the date of first publication is typically cast as a textual inference problem, estimating a date from linguistic features of the text alone (e.g., mentions of *airplanes* may date a book to after the turn of the 20th century; *thou* and *thee* offer evidence of composition before the 18th century, etc.). While it is reasonable to believe that memorized access to an entire book may lead to improved performance when predicting the date from a passage within it, another explanation is simply that GPT models are accessing encyclopedic knowledge about those works—i.e., identifying a passage as *Pride and Prejudice* and then accessing a learned fact about that work. For tasks that can be expressed as *facts about a text* (e.g., time of publication, genre, authorship), this presents a significant risk of contamination.

### 7.2 Predicting narrative time

Our second experiment predicts the amount of time that has passed in the same 250-word passage sampled above, using the conceptual framework of Underwood (2018), explored in the context of GPT-4 in Underwood (2023). Unlike the year prediction task above, which can potentially draw on encyclopedic knowledge that GPT-4 has memorized, this task requires reasoning directly about the information in the passage, as each passage has its own duration. Memorized information about a complete book could inform the prediction about all passages within it (e.g., through awareness of the general time scales present in a work, or the amount of dialogue within it), in addition to having exact knowledge of the passage itself.

| Model | missing, top 10% | missing, bot. 10% | $\rho$ | MAE, top 10% | MAE, bot. 10% |
|---|---|---|---|---|---|
| ChatGPT | 0.04 | 0.09 | -0.39 | 3.3 [0.1–12.3] | 29.8 [20.2-41.2] |
| GPT-4 | 0.00 | 0.00 | -0.37 | 0.6 [0.1–1.5] | 14.5 [9.5–20.8] |

Table 5: Date of first publication performance, with 95% bootstrap confidence intervals around MAE. Spearman $\rho$ across all data and all differences between top and bottom 10% within a model are significant ($p < 0.05$).

As above, our core question asks: do we see a difference in performance between books that GPT-4 has memorized and those that it has not? To assess this, we again identify the top and bottom decile of books by their GPT-4 name cloze accuracy. We manually annotate the amount of time passing in a 250-word passage from that book using the criteria outlined by Underwood (2018), including the examples provided to GPT-4 in Underwood (2023), blinding annotators to the identity of the book and which experimental condition it belongs to. We then pass those same passages through GPT-4 using the prompt in figure 4, adapted from Underwood (2023). We calculate accuracy as the Spearman $\rho$ between the predicted times and true times and calculate a 95% confidence interval over that measure using the bootstrap.

We find a large but not statistically significant difference between TOP$_{10}$ ($\rho = 0.50$ [0.25–0.72]) and BOT$_{10}$ ($\rho = 0.27$ [−0.02–0.59]), suggesting that GPT-4 may perform better on this task for books it has memorized, but not conclusive evidence that this is so. Expanding this same process to the top and bottom quintile sees no effect (TOP$_{20}$ ($\rho = 0.47$ [0.29–0.63]; BOT$_{20}$ ($\rho = 0.46$ [0.25–0.64]); if an effect exists, it is only among the most memorized books. This suggests caution; more research is required to assess these disparities.

## 8 Discussion

We carry out this work in anticipation of the appeal of ChatGPT and GPT-4 for both posing and answering questions in cultural analytics. We find that these models have memorized books, both in the public domain and in copyright, and the capacity for memorization is tied to a book's overall popularity on the web. This differential in memorization leads to differential in performance for downstream tasks, with better performance on popular books than on those not seen on the web. As we consider the use of these models for questions in cultural analytics, we see a number of issues worth considering.

**The virtues of open models.** This archaeology is required only because ChatGPT and GPT-4 are closed systems, and the underlying data on which they have been trained is unknown. While our work sheds light on the memorization behavior of these models, the true training data remains fundamentally unknowable outside of OpenAI. As others have pointed out with respect to these systems in particular, there are deep ethical and reproducibility concerns about the use of such closed models for scholarly research (Spirling, 2023). A preferred alternative is to embrace the development of more open, transparent models, including LLaMA (Touvron et al., 2023), OPT (Zhang et al., 2022) and BLOOM (Scao et al., 2022). While open-data models will not alleviate the disparities in performance we see by virtue of their openness (LLaMA is trained on books in the Pile, and all use information from the general web, which each allocate attention to popular works), they address the issue of data contamination since the works in the training data will be known; works in any evaluation data would then be able to be queried for membership (Marone and Van Durme, 2023).

**Popular texts are likely not good barometers of model performance.** A core concern about closed models with unknown training data is *test contamination*: data in an evaluation benchmark (providing an assessment of measurement validity) may be present in the training data, leading an assessment to be overconfident in a model's abilities. Our work here has shown that OpenAI models know about books in proportion to their popularity on the web, and that their performance on downstream tasks is tied to that popularity. When benchmarking the performance of these models on a new downstream task, it is risky to draw expectations of generalization from its performance on *Alice in Wonderland*, *Harry Potter*, *Pride and Prejudice*, and so on—it simply knows much more about these works than the long tail of literature (both in the public domain and in copyright). In this light, we hope this work can serve as the starting point for future work on the generalization ability of LLM

in the context of cultural analytics.

**Whose narratives?** At the core of questions surrounding the data used to train large language models is one of representation: whose language, and whose lived experiences mediated through that language, is captured in their knowledge of the world? While previous work has investigated this in terms of what varieties of English pass GPT-inspired content filters (Gururangan et al., 2022), we can ask the same question about the narrative experiences present in books: whose narratives inform the implicit knowledge of these models? Our work here has not only confirmed that ChatGPT and GPT-4 have memorized popular works, but also what kinds of narratives count as "popular"—in particular, works in the public domain published before 1928, along with contemporary science fiction and fantasy. Our work here does not investigate how this potential source of bias might become realized—e.g., how generated narratives are pushed to resemble text it has been trained on (Lucy and Bamman, 2021); how the values of the memorized texts are encoded to influence other behaviors, etc.—and so any discussion about it can only be speculative, but future work in this space can take the disparities we find in memorization as a starting point for this important work.

## 9 Conclusion

As research in cultural analytics is poised to be transformed by the affordances of large language models, we carry out this work to uncover the works of fiction that two popular models have memorized, in order to assess the risks of using closed models for research. In our finding that memorization affects performance on downstream tasks in disparate ways, closed data raises uncertainty about the conditions when a model will perform well and when it will fail; while our work tying memorization to popularity on the web offers a rough guide to assess the likelihood of memorization, it does not solve that fundamental challenge. Only open models with known training sources would do so.

Code and data to support this work can be found at `https://github.com/bamman-group/gpt4-books`.

## Limitations

The data behind ChatGPT and GPT-4 is fundamentally unknowable outside of OpenAI. Our work

carries out probabilistic inference to measure the familiarity of these models with a set of books, but the question of whether they truly exist within the training data of these models is not answerable. Mere familiarity, however, is enough to contaminate test data.

Our study has some limitations that future work could address. First, we focus on a case study of English, but the analytical method we employ is transferable to other languages (given digitized books written in those languages). We leave it to future work to quantify the extent of memorization in non-English literary content by large language models.

Second, the name cloze task is one operationalization to assess memorization of large language models. As shown by previous research (e.g., Lee et al., 2022), memorization can be much more acute with longer textual spans during training copied verbatim. We leave quantifying the full extent of memorization and data contamination of opaque models such as ChatGPT and GPT-4 for future work.

## Ethical considerations

Our study probes the degree to which ChatGPT and GPT-4 have memorized books, and what impact that memorization has on downstream research in cultural analytics. This work uses the OpenAI API to perform the name cloze task described above, and at no point do we access, or attempt to access, the true training data behind these models, or any underlying components of the systems.

## Acknowledgments

The research reported in this article was supported by funding from the National Science Foundation (IIS-1942591) and the National Endowment for the Humanities (HAA-271654-20).

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

# Appendix

## Author contributions

- Kent Chang: performed extrinsic analysis; annotated narrative time duration data; wrote paper.

- Mackenzie Cramer: performed human name cloze experiment; annotated narrative time duration data; researched self-publishing history of *Fifty Shades of Grey*.

- Sandeep Soni: performed the statistical analysis of predicted names; annotated narrative time duration data; wrote paper.

- David Bamman: conducted experiments, annotated narrative time duration data; wrote paper.

What was the year in which the following passage was written? You must make a prediction of a year, even if you are uncertain.

Example:

Input: He felt his face. 'I miss sneering at something I love. How we used to love to gather in the checker-tiled kitchen in front of the old boxy cathode-ray Sony whose reception was sensitive to airplanes and sneer at the commercial vapidity of broadcast stuff.'

Output: <year>1996</year>

Input: The multiplicity of its appeals—the perpetual surprise of its contrasts and resemblances! All these tricks and turns of the show were upon him with a spring as he descended the Casino steps and paused on the pavement at its doors. He had not been abroad for seven years—and what changes the renewed contact produced! If the central depths were untouched, hardly a pin-point of surface remained the same. And this was the very place to bring out the completeness of the renewal. The sublimities, the perpetuities, might have left him as he was: but this tent pitched for a day's revelry spread a roof of oblivion between himself and his fixed sky.

Output: <year>1905</year>

Input: He did not speak to Jack or G.H., nor they to him. He lit a cigarette with shaking fingers and watched the spinning billiard balls roll and gleam and clack over the green stretch of doth, dropping into holes after bounding to and fro from the rubber cushions. He felt impelled to say something to ease the swelling in his chest. Hurriedly, he flicked his cigarette into a spittoon and, with twin eddies of blue smoke jutting from his black nostrils, shouted hoarsely, "Jack, I betcha two bits you can't make it!" Jack did not answer; the ball shot straight across the table and vanished into a side pocket. "You would've lost," Jack said. "Too late now," Bigger said. "You wouldn't bet, so you lost." He spoke without looking. His entire body hungered for keen sensation, something exciting and violent to relieve the tautness. It was now ten minutes to three and Gus had not come. If Gus stayed away much longer, it would be too late. And Gus knew that. If they were going to do anything, it certainly ought to be done before folks started coming into the streets to buy their food for supper, and while the cop was down at the other end of the block.

Output:

Figure 3: Sample prompt for year of first publication prediction.

| GPT-4 | ChatGPT | BERT | Date | Author | Title |
|---|---|---|---|---|---|
| 0.76 | 0.43 | 0.00 | 1997 | J.K. Rowling | *Harry Potter and the Sorcerer's Stone* |
| 0.57 | 0.30 | 0.00 | 1949 | George Orwell | *1984* |
| 0.51 | 0.20 | 0.01 | 1954 | J.R.R. Tolkien | *The Fellowship of the Ring* |
| 0.49 | 0.16 | 0.13 | 2012 | E.L. James | *Fifty Shades of Grey* |
| 0.48 | 0.14 | 0.00 | 2008 | Suzanne Collins | *The Hunger Games* |
| 0.43 | 0.27 | 0.00 | 1954 | William Golding | *Lord of the Flies* |
| 0.43 | 0.17 | 0.00 | 1979 | Douglas Adams | *The Hitchhiker's Guide to the Galaxy* |
| 0.30 | 0.16 | 0.00 | 1959 | Chinua Achebe | *Things Fall Apart* |
| 0.28 | 0.12 | 0.00 | 1977 | J. R. R. Tolkien and Christopher Tolkien | *The Silmarillion* |
| 0.27 | 0.13 | 0.00 | 1953 | Ray Bradbury | *Fahrenheit 451* |
| 0.27 | 0.13 | 0.00 | 1996 | George R.R. Martin | *A Game of Thrones* |
| 0.26 | 0.05 | 0.01 | 2003 | Dan Brown | *The Da Vinci Code* |
| 0.26 | 0.08 | 0.00 | 1965 | Frank Herbert | *Dune* |
| 0.25 | 0.20 | 0.01 | 1937 | Zora Neale Hurston | *Their Eyes Were Watching God* |
| 0.25 | 0.14 | 0.00 | 1961 | Harper Lee | *To Kill a Mockingbird* |
| 0.24 | 0.03 | 0.03 | 1953 | Ian Fleming | *Casino Royale* |
| 0.22 | 0.13 | 0.00 | 1984 | William Gibson | *Neuromancer* |
| 0.20 | 0.10 | 0.00 | 1985 | Orson Scott Card | *Ender's Game* |
| 0.19 | 0.12 | 0.00 | 1932 | Aldous Huxley | *Brave New World* |
| 0.18 | 0.07 | 0.00 | 1937 | Margaret Mitchell | *Gone with the Wind* |
| 0.17 | 0.05 | 0.00 | 1968 | Philip K. Dick | *Do Androids Dream of Electric Sheep?* |
| 0.16 | 0.06 | 0.05 | 2009 | Dan Brown | *The Lost Symbol* |
| 0.15 | 0.04 | 0.04 | 2013 | Dan Brown | *Inferno* |
| 0.15 | 0.08 | 0.01 | 2014 | Veronica Roth | *Divergent* |
| 0.15 | 0.04 | 0.00 | 1940 | John Steinbeck | *The Grapes of Wrath* |
| 0.13 | 0.05 | 0.00 | 1983 | James Kahn | *Return of the Jedi* |
| 0.13 | 0.02 | 0.01 | 1928 | D. H. Lawrence | *Lady Chatterley's Lover* |
| 0.13 | 0.03 | 0.00 | 1977 | Alex Haley | *Roots* |
| 0.11 | 0.11 | 0.00 | 1961 | Irving Stone | *The Agony and the Ecstasy* |
| 0.11 | 0.01 | 0.03 | 1957 | Ian Fleming | *From Russia with Love* |
| 0.11 | 0.04 | 0.00 | 1962 | Madeleine L'Engle | *A Wrinkle in Time* |
| 0.11 | 0.04 | 0.01 | 1939 | Marjorie Kinnan Rawlings | *The Yearling* |
| 0.10 | 0.05 | 0.00 | 1975 | E. L. Doctorow | *Ragtime* |
| 0.10 | 0.05 | 0.00 | 1929 | Dashiell Hammett | *The Maltese Falcon* |
| 0.10 | 0.08 | 0.07 | 1991 | Diana Gabaldon | *Outlander* |
| 0.10 | 0.02 | 0.00 | 1989 | Kazuo Ishiguro | *The Remains of the Day* |
| 0.10 | 0.01 | 0.00 | 1983 | Alice Walker | *The Color Purple* |
| 0.09 | 0.02 | 0.00 | 1934 | Dorothy L. Sayers | *The Nine Tailors* |
| 0.09 | 0.03 | 0.00 | 1985 | Margaret Atwood | *The Handmaid's Tale* |
| 0.09 | 0.04 | 0.00 | 1988 | Toni Morrison | *Beloved* |
| 0.08 | 0.07 | 0.00 | 1982 | Joe Nazel | *Every Goodbye Ain't Gone* |
| 0.08 | 0.06 | 0.01 | 1984 | Helen Hooven Santmyer | *...And Ladies of the Club* |
| 0.08 | 0.02 | 0.00 | 2006 | Max Brooks | *World War Z* |
| 0.08 | 0.02 | 0.00 | 1993 | Irvine Welsh | *Trainspotting* |
| 0.08 | 0.03 | 0.00 | 1947 | Robert Penn Warren | *All the King's Men* |
| 0.07 | 0.02 | 0.00 | 1952 | Ralph Ellison | *Invisible Man* |
| 0.07 | 0.01 | 0.00 | 1951 | Isaac Asimov | *Foundation* |
| 0.07 | 0.01 | 0.00 | 1976 | Anne Rice | *Interview with the Vampire* |
| 0.07 | 0.04 | 0.02 | 1977 | Stephen King | *The Shining* |
| 0.07 | 0.01 | 0.00 | 1996 | Helen Fielding | *Bridget Jones's Diary* |

Table 6: Top 50 books in copyright (published after 1928) by GPT-4 name cloze accuracy.

Read the following passage of fiction. Then do five things.

1: Briefly summarize the passage.

2: Reason step by step to decide how much time is described in the passage. If the passage doesn't include any explicit reference to time, you can guess how much time the events described would have taken. Even description can imply the passage of time by mentioning the earlier history of people or buildings. But characters' references to the past or future in spoken dialogue should not count as time that passed in the scene. Report the time using units of years, weeks, days, hours, or minutes. Do not say zero or N/A.

3: If you described a range of possible times in step 2 take the midpoint of the range. Then multiply to convert the units into minutes.

4: Report only the number of minutes elapsed, which should match the number in step 3. Do not reply N/A.

5: Given the amount of speculation required in step 2, describe your certainty about the estimate–either high, moderate, or low.

Input:

TWENTY-FIVE It was raining again the next morning, a slanting gray rain like a swung curtain of crystal beads. I got up feeling sluggish and tired and stood looking out of the windows, with a dark harsh taste of Sternwoods still in my mouth. I was as empty of life as a scarecrow's pockets. I went out to the kitchenette and drank two cups of black coffee. You can have a hangover from other things than alcohol. I had one from women. Women made me sick. I shaved and showered and dressed and got my raincoat out and went downstairs and looked out of the front door. Across the street, a hundred feet up, a gray Plymouth sedan was parked. It was the same one that had tried to trail me around the day before, the same one that I had asked Eddie Mars about. There might be a cop in it, if a cop had that much time on his hands and wanted to waste it following me around. Or it might be a smoothie in the detective business trying to get a noseful of somebody else's case in order to chisel a way into it. Or it might be the Bishop of Bermuda disapproving of my night life.

Output:

1: A detective wakes up 'the next morning,' looks out a window for an undefined time, drinks (and presumably needs to make) two cups of coffee, then shaves and showers and gets dressed before stepping out his front door and seeing a car.
2: Making coffee, showering, and getting dressed take at least an hour. There's some ambiguity about whether to count the implicit reference to yesterday (since this is 'the next morning') as time elapsed in the passage, but let's say no, since yesterday is not actually described. So, an hour to 90 minutes.
3: 1.25 hours have elapsed. Multiplying by 60 minutes an hour that's 75 minutes.
4: 75 minutes.
5: Low confidence, because of ambiguity about a reference to the previous day.

Input:

CHAPTER I "TOM!" No answer. "TOM!" No answer. "What's gone with that boy, I wonder? You TOM!" No answer. The old lady pulled her spectacles down and looked over them about the room; then she put them up and looked out under them. She seldom or never looked *through* them for so small a thing as a boy; they were her state pair, the pride of her heart, and were built for "style," not service–she could have seen through a pair of stove-lids just as well. She looked perplexed for a moment, and then said, not fiercely, but still loud enough for the furniture to hear: "Well, I lay if I get hold of you I'll–" She did not finish, for by this time she was bending down and punching under the bed with the broom, and so she needed breath to punctuate the punches with. She resurrected nothing but the cat. "I never did see the beat of that boy!" She went to the open door and stood in it and looked out among the tomato vines and "jimpson" weeds that constituted the garden. No Tom. So she lifted up her voice at an angle calculated for distance and shouted: "Y-o-u-u TOM!" There was a slight noise behind her and she turned just in time to seize a small boy by the slack of his roundabout and arrest his flight.

Output:

1: An old lady calls for a boy named Tom, checks for him under the bed, goes to the open door and calls for him — then finally catches him.
2: The lady's actions are described minutely and seem rapid; they probably took two to four minutes. The lady also alludes to Tom's past bad behavior, but references to the past in dialogue should not count as time passing in the scene.
3: Three minutes have elapsed.
4: 3 minutes.
5: High confidence.

Input: [text]

Output:

Figure 4: Sample prompt for narrative time prediction, from Underwood (2023).