# OpenReview forum: "Speak, Memory: An Archaeology of Books Known to ChatGPT/GPT-4"
_EMNLP/2023/Conference — EMNLP 2023 Main_

### Official Review · Reviewer_swER · 2023-07-31

**Typos Grammar Style And Presentation Improvements:** N/A
**Soundness:** 4

**Excitement:**

4: Strong: This paper deepens the understanding of some phenomenon or lowers the barriers to an existing research direction.

**Missing References:**

N/A

**Paper Topic And Main Contributions:**

This paper presents an analysis of the books that are memorized in the training data of ChatGPT, GPT-4 and BERT large language models.
The main contribution of this work resides in its results, where these models have extensively memorized many books, especially popular sci-fi/fantasy novels but also including copyrighted bestsellers.

**Questions For The Authors:**

A) The name cloze evaluation, while creative, is an indirect measure of memorization. What do you think of direct methods like targeted probing?
B) You focus exclusively on fiction books in English. To what extent do you think findings could generalize to other text genres and languages?
C) The correlations shown between web prevalence and memorization are interesting but do not prove causation. Can you speculate on other factors that could also drive the degree of memorization seen?

**Reasons To Accept:**

-  Understanding memorization in large language models - that has significant implications for bias and validity in downstream tasks.
- The analysis is rigorous, evaluating hundreds of books across multiple models.
- The paper is well-written and presents the analysis clearly.

**Reasons To Reject:**

- The downstream tasks evaluating impact on performance are limited. More analysis of different tasks could strengthen the conclusions.
- The scale of the experiments, while a strength, required significant resources. This raises reproducibility concerns.

**Reproducibility:**

3: Could reproduce the results with some difficulty. The settings of parameters are underspecified or subjectively determined; the training/evaluation data are not widely available.

**Reviewer Confidence:**

4: Quite sure. I tried to check the important points carefully. It's unlikely, though conceivable, that I missed something that should affect my ratings.

---

> ### Author Rebuttal · Authors · 2023-08-29
>
> We are grateful for your recognition of the “significant implications for bias and validity in downstream tasks”. Our answers to your questions below:
>
> > A) The name cloze evaluation, while creative, is an indirect measure of memorization. What do you think of direct methods like targeted probing?
>
> Yes, targeted probing would be a more direct measure of memorization, but it would be most effective when we have hidden representations of the model for the inputs. To the best of our knowledge, the current OpenAI API does not give us access to the hidden representations.
>
> > B) You focus exclusively on fiction books in English. To what extent do you think findings could generalize to other text genres and languages?
>
> We believe that this is an important empirical question for future work. Our expectation is that similar training processes would yield similar facts of memorization (in which the more times a text is seen, the greater the chance it has been memorized); but as the size of data in different languages and genres vary in orders of magnitude with respect to English fiction, an important open question is many observations of a text are necessary to influence memorization (and downstream work in cultural analytics).
>
> > C) The correlations shown between web prevalence and memorization are interesting but do not prove causation. Can you speculate on other factors that could also drive the degree of memorization seen?
>
> This is a great point. It’s correct that we don’t specify a causal arrow in which memorization is directly explained by web prevalence; if we were to speculate, it’s likely that a hidden confounder explains them both: popularity.  Moby Dick, for example, is widely read; a.) many copies of it appear in university libraries, b.) multiple copies of it may appear in large-scale training datasets such as books3, c.) quotations of it may appear in unindexed academic journals, and d.) as we see, it appears in multiple places on the public web (both in snippets and in the form of full text).  All four of these factors are likely correlated with each other and commonly explained by the overall popularity of the text itself (concretely, by an unobservable quantity such as the number of times it has been read worldwide).  The correlation we see between web prevalence and memorization may be an indirect reflection of this latent causal graph; we will discuss this relationship (and the lack of direct causation between web prevalence and memorization) further in this paper.

---

### Official Review · Reviewer_gM9J · 2023-08-04

**Typos Grammar Style And Presentation Improvements:** 1) Please, provide more details about…
**Soundness:** 4

**Excitement:**

4: Strong: This paper deepens the understanding of some phenomenon or lowers the barriers to an existing research direction.

**Paper Topic And Main Contributions:**

The paper describes a data archaeology analysis investigating how closed large language models (ChatGPT/GPT-4) memorize books. This analysis is essential to understand which data are used to train these models. In particular, to investigate if data under copyright are used and to avoid using test data already present in the training (train-test leakage).

**Questions For The Authors:**

Q_A: LLaMA is not open source. Also, LLaMA 2 is not open source. Only the model weights are open, but the whole corpus used for the training and the code used for training are unavailable. Clarify this aspect.

**Reasons To Accept:**

1) Interesting and helpful analysis.
2) Reproducibility: code and data will be available. It is essential to release data generated by ChatGPT/GPT-4 since they can change.
3) The paper is clear and well-written.

**Reasons To Reject:**

1) Risk of speculations.
The authors provide an accurate methodology, but there is the risk that they already know what they find as result and design experiments to confirm what they want.
2) A task that involves fine-tuning should be helpful.
The authors design the prediction of publication year as a zero-shot task. However, they can use a portion of the data for fine-tuning the system and then evaluate the fine-tuned model on the other part. This can help to measure if memorization can also affect fine-tuning.
3) Limitation.
One of the limitations proposed by the author is the central issue of the paper. The paper fails to measure the LLMs' generalization ability in the case of memorization. We know that memorization has several drawbacks (copyright infringement, train-test leakage) and the analysis reported in the paper goes in these directions. However, the main question is: Can LLMs generalize when writing narrative texts? Also, humans improve their writing ability if they read books written by other people. The difference is that we do not memorize the whole passages from books.

**Reproducibility:**

5: Could easily reproduce the results.

**Reviewer Confidence:**

4: Quite sure. I tried to check the important points carefully. It's unlikely, though conceivable, that I missed something that should affect my ratings.

---

> ### Author Rebuttal · Authors · 2023-08-29
>
> Thank you for your thoughtful feedback, and we appreciate that you find the analysis in this paper clear and helpful.
>
> > The authors provide an accurate methodology, but there is the risk that they already know what they find as result and design experiments to confirm what they want.
>
> With regard to the risk of confirmation bias in the design of our experiments, we disagree for two reasons. First, we are not aware of any past research that attempted to quantify memorization in ChatGPT and GPT-4, especially in the context of books, so we could not have known our main finding *a priori*. Second, ChatGPT and GPT-4 have not released the details about their training data or protocols, so we could not have known the extent of copyrighted material that these models have been trained on before investigating ourselves.  Our work is designed to flesh out the contours of this memorization so that we, and others, are able to account for it in applying these models to cultural analytics; our work is descriptive and does not hinge on any specific finding of memorization.
>
> > A task that involves fine-tuning should be helpful.
>
> Yes, although we’d like to point out that §7 is first written with practitioners of cultural analytics in mind: What must we remember if we engage GPT-3.5/4 for inference tasks to bring to bear on literature and culture? The disparity in memorization leads to disparity in performance, which is important to note before we can draw any meaningful conclusion from the GPT-assisted analysis. The effects of fine-tuning on those tasks, while we agree could help us study memorization, are not what we look to investigate in this section. We’ll also note that fine-tuning with GPT-3.5/4 models was not possible at the time of writing. Thank you, we’ll clarify in the camera ready.
>
> > The paper fails to measure the LLMs' generalization ability in the case of memorization.
>
> It’s correct that we haven’t measured the generalization ability of these models accounting potentially for memorization, and we would agree that it warrants study, but we could only do that if we knew the boundaries between seen and unseen data for these models. In this light, as we allude to in the discussion, we see this paper as the starting point for this important future work. We’ll make this point explicit in the final version of the paper.
>
> > LLaMA is not open source. Also, LLaMA 2 is not open source. Only the model weights are open, but the whole corpus used for the training and the code used for training are unavailable.
>
> Thank you for calling our attention to our description of LLaMA; we will be sure to clarify in the camera-ready.
>
> > Please, provide more details about the human evaluation reported in line 207
>
> Yes, we will be including more details on the human evaluation of our name cloze task in the camera-ready.

---

### Official Review · Reviewer_2wkf · 2023-08-04

**Soundness:** 5

**Excitement:**

4: Strong: This paper deepens the understanding of some phenomenon or lowers the barriers to an existing research direction.

**Paper Topic And Main Contributions:**

This paper addresses the use of GPT-3.5 and GPT-4 and their potential for use in cultural analytics. The authors identify works memorised by the models by asking for predictions on un-guessable tokens, such as names. These estimates are then used to quantify the degree to which the openai models leverage this memoerisation in answering other tasks. The themes of the paper are copyright, memorisation, opaque models, and cultural analytics. The major contribution is the finding that there is significant impact from more-memorised works. The minor contribution is the use of the task as a determinant. I think this paper lies in the space between computationally-aided linguistic analysis and NLP engineering experiment.

**Questions For The Authors:**

1. is there any way to evaluate the perplexity or uniqueness of the segment itself, and its impact? “Stately, plump [MASK] came from the stairhead, bearing a bowl of lather on which a mirror and a razor lay crossed.” would itself quite an unusual construction, even without the mask.


**Reasons To Accept:**

The main topics of this work are timely, and hit on both fundamental questions of analysis (as seen in the discussion of cited works about the criticism of 'canon') with black-box tools, and the technical question of memorisation in LLMs. The paper has relevance beyond the core application area, and contributes significantly to the discussion of the state of the art. The methodology is sound, applies its statistical approach appropriately, and draws conclusions from the evidence.

**Reasons To Reject:**

I cannot think of any substantive reason to reject.

**Reproducibility:**

5: Could easily reproduce the results.

**Reviewer Confidence:**

4: Quite sure. I tried to check the important points carefully. It's unlikely, though conceivable, that I missed something that should affect my ratings.

---

> ### Author Rebuttal · Authors · 2023-08-29
>
> We appreciate your feedback and kind words!
>
> > Is there any way to evaluate the perplexity or uniqueness of the segment itself, and its impact?
>
> The short answer is no; calculating the perplexity of the prompt for the model requires the model to share the assigned probabilities to words or sequences. Unfortunately, ChatGPT and GPT-4 models currently don’t share them through their API.
>
> We agree this perplexity angle would be interesting. The perplexity of the prompt, as some research (e.g., Gonen et al. 2022) [1] has found, is linked to the model’s ability to generalize. The impact of having seen “unusual constructions” on perplexity is worth further investigation as well. As Liu et al. 2023 [2] noted, OpenAI has recently modified their model behavior to discourage reproducing copyrighted material, and it’s unclear whether this would figure in the calculation of perplexity for unique sentences without the mask.
>
> We will note this in the camera-ready version of our paper.
>
> [1] Hila Gonen, Srini Iyer, Terra Blevins, Noah A. Smith, Luke Zettlemoyer. “Demystifying Prompts in Language Models via Perplexity Estimation.” https://arxiv.org/abs/2212.04037
>
> [2] Yang Liu, Yuanshun Yao, Jean-Francois Ton, Xiaoying Zhang, Ruocheng Guo, Hao Cheng, Yegor Klochkov, Muhammad Faaiz Taufiq, Hang Li, “Trustworthy LLMs: a Survey and Guideline for Evaluating Large Language Models' Alignment.” https://arxiv.org/abs/2308.05374

---

### Meta-Review · Area_Chair_z73s · 2023-09-19

**Recommendation:** 5

**Metareview:**

The reviewers all agree that the paper draws attention to a timely and important question: which (potentially copyrighted) books appear in GPT-3.5/4's training data, and to what extend has GPT-3.5/4 memorized these works? Understanding the degree of memorization has important implications for researchers in the humanities who wish to use these models. The reviews also see broader applicability of this work beyond the humanities, contributing to a deeper understanding of the models. The reviewers agree that the experiments are well designed and implemented.

The reviewers raise several concerns, which the authors convincingly address in their rebuttal. Specifically, they explain why the name cloze  task is appropriate, why measuring the perplexity of the prompt is infeasible, and why fine tuning is less relevant in this situation.

---

### Decision · Program_Chairs · 2023-10-07

**Decision:**

Accept-Main

**Comment:**

The reviewers all agree that the paper draws attention to a timely and important question: which (potentially copyrighted) books appear in GPT-3.5/4's training data, and to what extend has GPT-3.5/4 memorized these works? Understanding the degree of memorization has important implications for researchers in the humanities who wish to use these models. The reviews also see broader applicability of this work beyond the humanities, contributing to a deeper understanding of the models. The reviewers agree that the experiments are well designed and implemented.

The reviewers raise several concerns, which the authors convincingly address in their rebuttal. Specifically, they explain why the name cloze  task is appropriate, why measuring the perplexity of the prompt is infeasible, and why fine tuning is less relevant in this situation.